# A Complete Variational Tracker

**Ryan Turner**
Northrop Grumman Corp.
ryan.turner@ngc.com

**Steven Bottone**
Northrop Grumman Corp.
steven.bottone@ngc.com

**Bhargav Avasarala**
Northrop Grumman Corp.
bhargav.avasarala@ngc.com

## Abstract

We introduce a novel probabilistic tracking algorithm that incorporates combinatorial data association constraints and model-based track management using variational Bayes. We use a Bethe entropy approximation to incorporate data association constraints that are often ignored in previous probabilistic tracking algorithms. Noteworthy aspects of our method include a model-based mechanism to replace heuristic logic typically used to initiate and destroy tracks, and an assignment posterior with linear computation cost in window length as opposed to the exponential scaling of previous MAP-based approaches. We demonstrate the applicability of our method on radar tracking and computer vision problems.

The field of *tracking* is broad and possesses many applications, particularly in radar/sonar [1], robotics [14], and computer vision [3]. Consider the following problem: A radar is tracking a flying object, referred to as a *target*, using measurements of range, bearing, and elevation; it may also have Doppler measurements of radial velocity. We would like to construct a *track* which estimates the trajectory of the object over time. The Kalman filter [16], or a more general state space model, is used to filter out measurement errors. The key difference between tracking and filtering is the presence of *clutter* (noise measurements) and *missed detections* of true objects. We must determine which measurement to "plug in" to the filter before applying it; this is known as *data association*. Additionally complicating the situation is that we may be in a *multi-target* tracking scenario in which there are multiple objects to track and we do not know which measurement originated from which object.

There is a large body of work on tracking algorithms given its standing as a long-posed and important problem. Algorithms vary primarily on their approach to data association. The dominant approach uses a *sliding window* MAP estimate of the measurement-to-track *assignment*, in particular the multiple hypothesis tracker (MHT) [1]. In the standard MHT, at every *frame* the algorithm finds the most likely matching of measurements to tracks, in the form of an *assignment matrix*, under a one-to-one constraint (see Figure 1). One track can only result in one measurement, and vice versa, which we refer to as *framing constraints*. As is typical in MAP estimation, once an assignment is determined, the filters are updated and the tracker proceeds as if these assignments were known to be correct. The one-to-one constraint makes MAP estimation a bipartite matching task where algorithms exist to solve it exactly in polynomial time in the number of tracks $N_T$ [15]. However, the multi-frame MHT finds the joint MAP assignment over multiple frames, in which case the assignment problem is known to be NP-hard, although good approximate solvers exist [20].

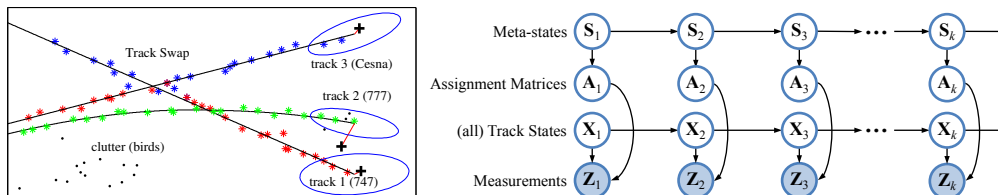

Figure 1: Simple scenario with a track swap: filtered state estimates $*$, associated measurements $+$, and clutter $\cdot$; and corresponding graphical model. Note that $\mathbf{X}_k$ is a matrix since it contains state vectors for all three tracks.

Despite the complexity of the MHT, it only finds a sliding window MAP estimate of measurement-to-track assignments. If a clutter measurement is by chance associated with a track for the duration of a window then the tracker will assume with certainty that the measurement originated from that track, and never reconsider despite all future evidence to the contrary. If multiple clutter (or otherwise incorrect) measurements are associated with a track, then it may veer "off into space" and result in *spurious* tracks. Likewise, an endemic problem in tracking is the issue of *track swaps*, where two trajectories can cross and get mixed up as shown in Figure 1. Alternatives to the MAP approach include the probabilistic MHT (PMHT) [9, Ch. 4] and probabilistic data association (PDA). However, the PMHT drops the one-to-one constraint in data association and the PDA only allows for a single target. This led to the development of the joint PDA (JPDA) algorithm for multiple targets, which utilizes heuristic calculations of the assignment weights and does not scale to multiple frame assignment. Particle filter implementations of the JPDA have tried to alleviate these issues, but they have not been adopted into real-time systems due to their inefficiency and lack of robustness. The probability hypothesis density (PHD) filter [19] addresses many of these issues, but only estimates the intensity of objects and does not model full trajectories; this is undesirable since the identity of an object is required for many applications including the examples in this paper.

Lázaro-Gredilla et al. [18] made the first attempt at a variational Bayes (VB) tracker. In their approach every trajectory follows a Gaussian process (GP); measurements are thus modeled by a mixture of GPs. We develop additional VB machinery to retain the framing constraints, which are dropped in Lázaro-Gredilla et al. [18] despite being viewed as important in many systems. Secondly, our algorithm utilizes a state space approach (e.g. Kalman filters) to model tracks, providing *linear* rather than *cubic* time complexity in track length. Hartikainen and Särkkä [11] showed by an equivalence that there is little loss of modeling flexibility by taking a state space approach over GPs.

Most novel tracking algorithms neglect the critical issue of *track management*. Many tracking algorithms unrealistically assume that the number of tracks $N_T$ is known a priori and fixed. Additional "wrapper logic" is placed around the trackers to initiate and destroy tracks. This logic involves many heuristics such as $M$-of-$N$ logic [1, Ch. 3]. Our method replaces these heuristics in a model-based manner to make significant performance gains. We call our method a *complete* variational tracker as it simultaneously does inference for track management, data association, and state estimation.

The outline of the paper is as follows: We first describe the full joint probability distribution of the tracking problem in Section 1. This includes how to solve the track management problem by augmenting tracks with an active/dormant state to address the issue of an unknown number of tracks. By studying the full joint we develop a new *conjugate prior* on assignment matrices in Section 2. Using this new formulation we develop a variational algorithm for estimating the measurement-to-track assignments and track states in Section 3. To retain the framing constraints and efficiently scale in tracks and measurements, we modify the variational lower bound in Section 4 using a Bethe entropy approximation. This results in a loopy belief propagation (BP) algorithm being used as a subroutine in our method. In Sections 5–6 we show the improvements our method makes on a difficult radar tracking example and a real data computer vision problem in sports.

Our paper presents the following novel contributions: First, we develop the first efficient deterministic approximate inference algorithm for solving the full tracking problem, which includes the framing constraints and track management. The most important observation is that the VB assignment posterior has an *induced factorization* over time with regard to assignment matrices. Therefore, the computational cost of our variational approach is *linear* in window length as opposed to the *exponential* cost of the MAP approach. The most astounding aspect is that by introducing a weaker approximation (VB factorization vs MAP) we lower the computational cost from exponential to linear; this is a truly rare and noteworthy example. Second, in the process, we develop new approximate inference methods on assignment matrices and a new conjugate assignment prior (CAP). We believe these methods have much larger applicability beyond our current tracking algorithm. Third, we develop a process to handle the track management problem in a model-based way.

# 1   Model Setup for the Tracking Problem

In this section we describe the full model used in the tracking problem and develop an unambiguous notation. At each time step $k \in \mathbb{N}_1$, known as a frame, we observe $N_Z(k) \in \mathbb{N}_0$ measurements, in a matrix $\mathbf{Z}_k = \{\mathbf{z}_{j,k}\}_{j=1}^{N_Z(k)}$, from both real targets and clutter (spurious measurements). In the

radar example $\mathbf{z}_{j,k} \in \mathcal{Z}$ is a vector of position measurements in $\mathbb{R}^3$. In data association we estimate the assignment matrices $\mathbf{A}$, where $A_{ij} = 1$ if and only if track $i$ is associated with measurement $j$. Recall that each track is associated with *at most* one measurement, and vice versa, implying:

$$\sum_{i=0}^{N_T} A_{ij} = 1\,, \quad j \in 1{:}N_Z\,, \quad \sum_{j=0}^{N_Z} A_{ij} = 1\,, \quad i \in 1{:}N_T\,, \quad A_{00} = 0\,. \tag{1}$$

The zero indices of $\mathbf{A} \in \{0,1\}^{N_T+1 \times N_Z+1}$ are the "dummy row" and "dummy column" to represent the assignment of a measurement to clutter and the assignment of a track to a missed detection.

**Distribution on Assignments**  Although not explicitly stated in the literature, a careful examination of the cost functions used in the MAP optimization in MHT yields a particular and intuitive prior on the assignment matrices. The number of tracks $N_T$ is assumed known a priori and $N_Z$ is random. The corresponding generative process on assignment matrices is as follows: 1) Start with a one-to-one mapping from measurements to tracks: $\mathbf{A} \leftarrow \mathbf{I}_{N_T \times N_T}$. 2) Each track is observed with probability $P_D \in [0,1]^{N_T}$. Only keep the columns of detected tracks: $\mathbf{A} \leftarrow \mathbf{A}(\cdot, \mathbf{d})$, $d_i \sim \text{Bernoulli}(P_D(i))$. 3) Sample a Poisson number of clutter measurements (columns): $\mathbf{A} \leftarrow [\mathbf{A}, \mathbf{0}_{N_T \times N_c}]$, $N_c \sim \text{Poisson}(\lambda)$. 4) Use a random permutation vector $\boldsymbol{\pi}$ to make the measurement order arbitrary: $\mathbf{A} \leftarrow \mathbf{A}(\cdot, \boldsymbol{\pi})$. 5) Append a dummy row and column on $\mathbf{A}$ to satisfy the summation constraints (1). This process gives the following normalized prior on assignments:

$$P(\mathbf{A}|P_D) = \lambda^{N_c} \exp(-\lambda)/N_Z! \prod_{i=1}^{N_T} P_D(i)^{d_i}(1 - P_D(i))^{1-d_i}\,. \tag{2}$$

Note that the detections $\mathbf{d}$, $N_Z$, and clutter measurement count $N_c$ are deterministic functions of $\mathbf{A}$.

**Track Model**  We utilize a state space formulation over $K$ time steps. The latent states $\mathbf{x}_{1:K} \in \mathcal{X}^K$ follow a Markov process, while the measurements $\mathbf{z}_{1:K} \in \mathcal{Z}^K$ are iid conditional on the track state:

$$p(\mathbf{z}_{1:K}, \mathbf{x}_{1:K}) = p(\mathbf{x}_1) \prod_{k=2}^{K} p(\mathbf{x}_k|\mathbf{x}_{k-1}) \prod_{k=1}^{K} p(\mathbf{z}_k|\mathbf{x}_k)\,, \tag{3}$$

where we have dropped the track and measurements indices $i$ and $j$. Although more general models are possible, within this paper each track independently follows a linear system (i.e. Kalman filter):

$$p(\mathbf{x}_k|\mathbf{x}_{k-1}) = \mathcal{N}(\mathbf{x}_k|\mathbf{F}\mathbf{x}_{k-1}, \mathbf{Q})\,, \quad p(\mathbf{z}_k|\mathbf{x}_k) = \mathcal{N}(\mathbf{z}_k|\mathbf{H}\mathbf{x}_k, \mathbf{R})\,. \tag{4}$$

**Track Meta-states**  We address the track management problem by augmenting track states with a two-state Markov model with an active/dormant meta-state $\mathbf{s}_k$ in a 1-of-$N$ encoding:

$$P(\mathbf{s}_{1:K}) = P(\mathbf{s}_1) \prod_{k=2}^{K} P(\mathbf{s}_k|\mathbf{s}_{k-1})\,, \quad \mathbf{s}_k \in \{0,1\}^{N_S}\,. \tag{5}$$

This effectively allows us to handle an unknown number of tracks by making $N_T$ arbitrarily large; $P_D$ is now a function of $\mathbf{s}$ with a very small $P_D$ in the dormant state and a larger $P_D$ in the active state. Extensions with a larger number of states $N_S$ are easily implementable. We refer to the collection of track meta-states over all tracks at frame $k$ as $\mathbf{S}_k := \{\mathbf{s}_{i,k}\}_{i=1}^{N_T}$; likewise, $\mathbf{X}_k := \{\mathbf{x}_{i,k}\}_{i=1}^{N_T}$.

**Full Model**  We combine the assignment process and track models to get the full model joint:

$$p(\mathbf{Z}_{1:K}, \mathbf{X}_{1:K}, \mathbf{A}_{1:K}, \mathbf{S}_{1:K}) = \prod_{k=1}^{K} p(\mathbf{Z}_k|\mathbf{X}_k, \mathbf{A}_k)p(\mathbf{X}_k|\mathbf{X}_{k-1})P(\mathbf{S}_k|\mathbf{S}_{k-1})P(\mathbf{A}_k|\mathbf{S}_k) \tag{6}$$

$$= \prod_{k=1}^{K} P(\mathbf{A}_k|\mathbf{S}_k) \cdot \prod_{i=1}^{N_T} p(\mathbf{x}_{i,k}|\mathbf{x}_{i,k-1})P(\mathbf{s}_{i,k}|\mathbf{s}_{i,k-1}) \cdot \prod_{j=1}^{N_Z(k)} p_0(\mathbf{z}_{j,k})^{A_{0j}^k} \prod_{i=1}^{N_T} p(\mathbf{z}_{j,k}|\mathbf{x}_{i,k}, A_{ij}^k = 1)^{A_{ij}^k}\,,$$

where $p_0$ is the *clutter distribution*, which is often a uniform distribution. The traditional goal in tracking is to compute $p(\mathbf{X}_k|\mathbf{Z}_{1:k})$, the exact computation of which is intractable due to the "combinatorial explosion" in summing out the assignments $\mathbf{A}_{1:k}$. The MHT MAP-based approach tackles this with $P(\mathbf{A}_{k_1:k_2}|\mathbf{Z}_{1:k}) \approx \mathbb{I}\{\mathbf{A}_{k_1:k_2} = \hat{\mathbf{A}}_{k_1:k_2}\}$ for a sliding window $w = k_2 - k_1 + 1$. Clearly an approximation is needed, but we show how to do much better than the MAP approach of the MHT. This motivates the next section where we derive a conjugate prior on the assignments $\mathbf{A}_{1:k}$, which is useful for improving upon MAP; and we cast (2) as a special case of this distribution.

## 2 The Conjugate Assignment Prior

Given that we must compute the posterior $P(\mathbf{A}|\mathbf{Z})$,[1] it is natural to ask what conjugate priors on $\mathbf{A}$ are possible. Deriving approximate inference procedures is often greatly simplified if the prior on the parameters is conjugate to the *complete data* likelihood: $p(\mathbf{Z}, \mathbf{X}|\mathbf{A})$ [2]. We follow the standard procedure for deriving the conjugate prior for an exponential family (EF) complete likelihood:

$$p(\mathbf{Z}, \mathbf{X}|\mathbf{A}) = \prod_{j=1}^{N_Z} p_0(\mathbf{z}_j)^{A_{0j}} \prod_{i=1}^{N_T} p(\mathbf{z}_j|\mathbf{x}_i, A_{ij}=1)^{A_{ij}} \prod_{i=1}^{N_T} p(\mathbf{x}_i) = \prod_{i=1}^{N_T} p(\mathbf{x}_i) \exp(\mathbf{1}^\top (\mathbf{A} \odot \mathbf{L})\mathbf{1}),$$

$$\mathbf{L}_{ij} := \log p(\mathbf{z}_j|\mathbf{x}_i, A_{ij}=1), \quad \mathbf{L}_{i0} := 0, \quad \mathbf{L}_{0j} := \log p_0(\mathbf{z}_j), \tag{7}$$

where we have introduced the matrix $\mathbf{L} \in \mathbb{R}^{N_T+1 \times N_Z+1}$ to represent log likelihood contributions from various assignments. Therefore, we have the following EF quantities [4, Ch. 2.4]: base measure $h(\mathbf{Z}, \mathbf{X}) = \prod_{i=1}^{N_T} p(\mathbf{x}_i)$, partition function $g(\mathbf{A}) = 1$, natural parameters $\eta(\mathbf{A}) = \text{vec}\,\mathbf{A}$, and sufficient statistics $T(\mathbf{Z}, \mathbf{X}) = \text{vec}\,\mathbf{L}$. This implies the conjugate assignments prior (CAP) for $P(\mathbf{A}|\boldsymbol{\chi})$:

$$\text{CAP}(\mathbf{A}|\boldsymbol{\chi}) := \mathcal{Z}(\boldsymbol{\chi})^{-1} \mathbb{I}\{\mathbf{A} \in \mathcal{A}\} \exp(\mathbf{1}^\top(\boldsymbol{\chi} \odot \mathbf{A})\mathbf{1}), \ \mathcal{Z}(\boldsymbol{\chi}) := \sum_{\mathbf{A} \in \mathcal{A}} \exp(\mathbf{1}^\top(\boldsymbol{\chi} \odot \mathbf{A})\mathbf{1}), \tag{8}$$

where $\mathcal{A}$ is the set of all assignment matrices that obey the one-to-one constraints (1). Note that $\boldsymbol{\chi}$ is a function of the track meta-states $\mathbf{S}$. We recover the assignment prior of (2) in the form of the CAP distribution (8) via the following parameter settings, with $\sigma(\cdot)$ denoting the logistic,

$$\chi_{ij} = \log\left(\frac{P_D(i)}{(1-P_D(i))\lambda}\right) = \sigma^{-1}(P_D(i)) - \log \lambda, \ i \in 1{:}N_T, \ j \in 1{:}N_Z, \ \chi_{0j} = \chi_{i0} = 0. \tag{9}$$

Due to the symmetries in the prior of (9) we can analytically normalize (8) in this special case:

$$\mathcal{Z}(\boldsymbol{\chi})^{-1} = P(\mathbf{A}_{1:N_T, 1:N_Z} = \mathbf{0}) = \text{Poisson}(N_Z|\lambda) \prod_{i=1}^{N_T}(1 - P_D(i)). \tag{10}$$

Given that the dummy row and columns of $\boldsymbol{\chi}$ are zero in (9), equation (10) is clearly the only way to get (8) to match (2) for the $\mathbf{0}$ assignment case.

Although the conjugate prior (8) allows us to "compute" the posterior, $\boldsymbol{\chi}_{\text{posterior}} = \boldsymbol{\chi}_{\text{prior}} + \mathbf{L}$, computing $\mathbb{E}[\mathbf{A}]$ or $\mathcal{Z}(\boldsymbol{\chi})$ remains difficult in general. This will cause problems in Section 3, but be ameliorated in Section 4 by a slight modification of the variational objective.

One insight into the partition function $\mathcal{Z}(\boldsymbol{\chi})$ is that if we slightly change the constraints in $\mathcal{A}$ so that all the rows and columns must sum to one, i.e. we do not use a dummy row or column and $\mathcal{A}$ becomes the set of *permutation* matrices, then $\mathcal{Z}(\boldsymbol{\chi})$ is equal to the matrix *permanent* of $\exp(\boldsymbol{\chi})$, which is #P-complete to compute [24]. Although the matrix permanent is #P-complete, accurate and computationally efficient approximations exist, some based on *belief propagation* [25; 17].

## 3 Variational Formulation

As explained in Section 1, exact inference on the full model in (6) is intractable, and as promised we show how to perform better inference than the existing solution of sliding window MAP. Our variational tracker enforces the factorization constraint that the posterior factorizes across assignment matrices and latent track states:

$$p(\mathbf{A}_{1:K}, \mathbf{X}_{1:K}, \mathbf{S}_{1:K}|\mathbf{Z}_{1:K}) \approx q(\mathbf{A}_{1:K}, \mathbf{X}_{1:K}, \mathbf{S}_{1:K}) = q(\mathbf{A}_{1:K})q(\mathbf{X}_{1:K}, \mathbf{S}_{1:K}). \tag{11}$$

In some sense we can think of $\mathbf{A}$ as the "parameters" with $\mathbf{X}$ and $\mathbf{S}$ as the "latent variables" and use the common variational practice of factorizing these two groups of variables. This gives the variational lower bound $\mathcal{L}(q)$:

$$\mathcal{L}(q) = \mathbb{E}_q[\log p(\mathbf{Z}_{1:K}, \mathbf{X}_{1:K}, \mathbf{A}_{1:K}, \mathbf{S}_{1:K})] + \text{H}[q(\mathbf{X}_{1:K}, \mathbf{S}_{1:K})] + \text{H}[q(\mathbf{A}_{1:K})], \tag{12}$$

where $\mathrm{H}[\cdot]$ denotes the Shannon entropy. From inspecting the VB lower bound (12) and (6) we arrive at the following *induced factorizations* without forcing further factorization upon (11):

$$q(\mathbf{A}_{1:K}) = \prod_{k=1}^{K} q(\mathbf{A}_k), \quad q(\mathbf{X}_{1:K}, \mathbf{S}_{1:K}) = \prod_{i=1}^{N_T} q(\mathbf{x}_{i,\cdot}) q(\mathbf{s}_{i,\cdot}). \tag{13}$$

In other words, the approximate posterior on assignment matrices factorizes across time; and the approximate posterior on latent states factorizes across tracks.

**State Posterior Update** Based on the induced factorizations in (13) we derive the updates for the track states $\mathbf{x}_{i,\cdot}$ and meta-states $\mathbf{s}_{i,\cdot}$ separately. Additionally, we derive the updates for each track separately. We begin with the variational updates for $q(\mathbf{x}_{i,\cdot})$ using the standard VB update rules [4, Ch. 10] and (6), denoting equality to an additive constant with $\overset{c}{=}$,

$$\log q(\mathbf{x}_{i,\cdot}) \overset{c}{=} \log p(\mathbf{x}_{i,\cdot}) + \sum_{k=1}^{K} \sum_{j=1}^{N_Z(k)} \mathbb{E}[A_{ij}^k] \log \mathcal{N}(\mathbf{z}_{j,k} | \mathbf{H}\mathbf{x}_{i,k}, \mathbf{R}) \tag{14}$$

$$\implies q(\mathbf{x}_{i,\cdot}) \propto p(\mathbf{x}_{i,\cdot}) \prod_{k=1}^{K} \prod_{j=1}^{N_Z(k)} \mathcal{N}(\mathbf{z}_{j,k} | \mathbf{H}\mathbf{x}_{i,k}, \mathbf{R}/\mathbb{E}[A_{ij}^k]). \tag{15}$$

Using the standard product of Gaussians formula [6] this is proportional to

$$q(\mathbf{x}_{i,\cdot}) \propto p(\mathbf{x}_{i,\cdot}) \prod_{k=1}^{K} \mathcal{N}(\tilde{\mathbf{z}}_{i,k} | \mathbf{H}\mathbf{x}_{i,k}, \mathbf{R}/\mathbb{E}[d_{i,k}]), \quad \tilde{\mathbf{z}}_{i,k} := \frac{1}{\mathbb{E}[d_{i,k}]} \sum_{j=1}^{N_Z} \mathbb{E}[A_{ij}^k] \mathbf{z}_{j,k}, \tag{16}$$

and recall that $\mathbb{E}[d_{i,k}] = 1 - \mathbb{E}[A_{i0}^k] = \sum_{j=1}^{N_Z} \mathbb{E}[A_{ij}^k]$. The form of the posterior $q(\mathbf{x}_{i,\cdot})$ is equivalent to a linear dynamical system with pseudo-measurements $\tilde{\mathbf{z}}_{i,k}$ and non-stationary measurement covariance $\mathbf{R}/\mathbb{E}[d_{i,k}]$. Therefore, $q(\mathbf{x}_{i,\cdot})$ is simply implemented using a *Kalman smoother* [22].

**Meta-state Posterior Update** We next consider the posterior on the track meta-states:

$$\log q(\mathbf{s}_{i,\cdot}) \overset{c}{=} \log P(\mathbf{s}_{i,\cdot}) + \sum_{k=1}^{K} \mathbb{E}_{q(\mathbf{A}_k)}[\log P(\mathbf{A}_k | \mathbf{S}_k)] \overset{c}{=} \log P(\mathbf{s}_{i,\cdot}) + \sum_{k=1}^{K} \mathbf{s}_{i,k}^{\top} \boldsymbol{\ell}_{i,k}, \tag{17}$$

$$\boldsymbol{\ell}_{i,k}(s) := \mathbb{E}[d_{i,k}] \log(P_D(s)) + (1 - \mathbb{E}[d_{i,k}]) \log(1 - P_D(s)), \quad s \in 1{:}N_S \tag{18}$$

$$\implies q(\mathbf{s}_{i,\cdot}) \propto P(\mathbf{s}_{i,\cdot}) \prod_{k=1}^{K} \exp(\mathbf{s}_{i,k}^{\top} \boldsymbol{\ell}_{i,k}), \tag{19}$$

where (18) follows from (2). If $P(\mathbf{s}_{i,\cdot})$ follows a Markov chain then the form for $q(\mathbf{s}_{i,\cdot})$ is the same as a hidden Markov model (HMM) with emission log likelihoods $\boldsymbol{\ell}_{i,k} \in [\mathbb{R}^-]^{N_S}$. Therefore, the meta-state posterior $q(\mathbf{s}_{i,\cdot})$ update is implemented using the *forward-backward* algorithm [21].

Like the MHT, our algorithm also works in an online fashion using a (much larger) sliding window.

**Assignment Matrix Update** The reader can verify using (7)–(9) that the exact updates under the lower bound $\mathcal{L}(q)$ (12) yields a product of CAP distributions:

$$q(\mathbf{A}_{1:K}) = \prod_{k=1}^{K} \mathrm{CAP}(\mathbf{A}_k | \mathbb{E}_{q(\mathbf{X}_k)}[\mathbf{L}_k] + \mathbb{E}_{q(\mathbf{S}_k)}[\boldsymbol{\chi}_k]). \tag{20}$$

This poses a challenging problem, as the state posterior updates of (16) and (19) require $\mathbb{E}_{q(\mathbf{A}_k)}[\mathbf{A}_k]$; since $q(\mathbf{A}_k)$ is a CAP distribution we know from Section 2 its expectation is difficult to compute.

## 4 The Assignment Matrix Update Equations

In this section we modify the variational lower bound (12) to obtain a tractable algorithm. The resulting algorithm uses loopy belief propagation to compute $\mathbb{E}_{q(\mathbf{A}_k)}[\mathbf{A}_k]$ for use in (16) and (19).

We first note that the CAP distribution (8) is naturally represented as a factor graph:

$$\text{CAP}(\mathbf{A}|\boldsymbol{\chi}) \propto \prod_{i=1}^{N_T} f_i^R(\mathbf{A}_{i\cdot}) \prod_{j=1}^{N_Z} f_j^C(\mathbf{A}_{\cdot j}) \prod_{i=0}^{N_T} \prod_{j=0}^{N_Z} f_{ij}^S(A_{ij}) \,, \tag{21}$$

with $f_i^R(\mathbf{v}) := \mathbb{I}\{\sum_{j=0}^{N_Z} \mathbf{v}_j = 1\}$ ($R$ for row factors), $f_j^C(\mathbf{v}) := \mathbb{I}\{\sum_{i=0}^{N_T} \mathbf{v}_i = 1\}$ ($C$ for column factors), and $f_{ij}^S(v) := \exp(\chi_{ij} v)$. We use reparametrization methods (see [10]) to convert (21) to a pairwise factor graph, where derivation of the *Bethe free energy* is easier. The *Bethe entropy* is:

$$\text{H}_\beta[q(\mathbf{A})] := \sum_{i=1}^{N_T} \sum_{j=0}^{N_Z} \text{H}[q(\mathbf{r}_i, A_{ij})] + \sum_{j=1}^{N_Z} \sum_{i=0}^{N_T} \text{H}[q(\mathbf{c}_j, A_{ij})]$$

$$- \sum_{i=1}^{N_T} N_Z \text{H}[q(\mathbf{r}_i)] - \sum_{j=1}^{N_Z} N_T \text{H}[q(\mathbf{c}_j)] - \sum_{i=1}^{N_T} \sum_{j=1}^{N_Z} \text{H}[q(A_{ij})] \tag{22}$$

$$= \sum_{i=1}^{N_T} \text{H}[q(\mathbf{A}_{i\cdot})] + \sum_{j=1}^{N_Z} \text{H}[q(\mathbf{A}_{\cdot j})] - \sum_{i=1}^{N_T} \sum_{j=1}^{N_Z} \text{H}[q(A_{ij})] \,, \tag{23}$$

where the pairwise conversion used constrained auxiliary variables $\mathbf{r}_i := \mathbf{A}_{i\cdot}$ and $\mathbf{c}_j := \mathbf{A}_{\cdot j}$; and used the implied relations $\text{H}[q(\mathbf{r}_i, A_{ij})] = \text{H}[q(\mathbf{r}_i)] + \text{H}[q(A_{ij}|\mathbf{r}_i)] = \text{H}[q(\mathbf{r}_i)] = \text{H}[q(\mathbf{A}_{i\cdot})]$.

We define an altered variational lower bound $\mathcal{L}_\beta(q)$, which merely replaces the entropy $\text{H}[q(\mathbf{A}_k)]$ with $\text{H}_\beta[q(\mathbf{A}_k)]$.[2] Note that $\mathcal{L}_\beta(q) \overset{c}{=} \mathcal{L}(q)$ with respect to $q(\mathbf{X}_{1:K}, \mathbf{S}_{1:K})$, which implies the state posterior updates under the old bound $\mathcal{L}(q)$ in (16) and (19) remain unchanged with the new bound $\mathcal{L}_\beta(q)$. To get the new update equations for $q(\mathbf{A}_k)$ we examine $\mathcal{L}_\beta(q)$ in terms of $q(\mathbf{A}_{1:K})$:

$$\mathcal{L}_\beta(q) \overset{c}{=} \mathbb{E}_q[\log p(\mathbf{Z}_{1:K}|\mathbf{X}_{1:K}, \mathbf{A}_{1:K})] + \mathbb{E}_q[\log P(\mathbf{A}_{1:K}|\mathbf{S}_{1:K})] + \sum_{k=1}^{K} \text{H}_\beta[q(\mathbf{A}_k)] \tag{24}$$

$$\overset{c}{=} \sum_{k=1}^{K} \mathbb{E}_{q(\mathbf{A}_k)}[\mathbf{1}^\top (\mathbf{A}_k \odot (\mathbb{E}_{q(\mathbf{X}_k)}[\mathbf{L}_k] + \mathbb{E}_{q(\mathbf{S}_k)}[\boldsymbol{\chi}_k]))\mathbf{1}] + \sum_{k=1}^{K} \text{H}_\beta[q(\mathbf{A}_k)] \tag{25}$$

$$\overset{c}{=} \sum_{k=1}^{K} \mathbb{E}_{q(\mathbf{A}_k)}[\log \text{CAP}(\mathbf{A}_k|\mathbb{E}_{q(\mathbf{X}_k)}[\mathbf{L}_k] + \mathbb{E}_{q(\mathbf{S}_k)}[\boldsymbol{\chi}_k])] + \text{H}_\beta[q(\mathbf{A}_k)] \,. \tag{26}$$

This corresponds to the Bethe free energy of the factor graph described in (21), with $\mathbb{E}[\mathbf{L}_k] + \mathbb{E}[\boldsymbol{\chi}_k]$ as the CAP parameter [26; 12]. Therefore, we can compute $\mathbb{E}[\mathbf{A}_k]$ using loopy belief propagation.

**Loopy BP Derivation**  We define the key (row/column) quantities for the belief propagation:

$$\mu_{ij}^R := \text{msg}_{f_i^R \to A_{ij}} \,, \quad \mu_{ij}^C := \text{msg}_{f_j^C \to A_{ij}} \,, \quad \nu_{ij}^R := \text{msg}_{A_{ij} \to f_i^R} \,, \quad \nu_{ij}^C := \text{msg}_{A_{ij} \to f_j^C} \,,$$

where all messages form functions in $\{0,1\} \to \mathbb{R}^+$. Using the standard rules of BP we derive:

$$\nu_{ij}^R(x) = \mu_{ij}^C(x) f_{ij}^S(x) \,, \quad \mu_{ij}^R(1) = \prod_{k\neq j} \nu_{ik}^R(0) \,, \quad \mu_{ij}^R(0) = \sum_{l\neq j} \nu_{il}^R(1) \prod_{k\neq j,l} \nu_{ik}^R(0) \,, \tag{27}$$

where we have exploited that there is only one nonzero value in the row $\mathbf{A}_{i,\cdot}$. Notice that

$$\mu_{ij}^R(1) = \prod_{k=0}^{N_Z} \nu_{ik}^R(0)/\nu_{ij}^R(0) \implies \tilde{\mu}_{ij}^R := \frac{\mu_{ij}^R(0)}{\mu_{ij}^R(1)} = \sum_{l=0}^{N_Z} \frac{\nu_{il}^R(1)}{\nu_{il}^R(0)} - \frac{\nu_{ij}^R(1)}{\nu_{ij}^R(0)} \in \mathbb{R}^+ \,, \tag{28}$$

where we have pulled $\mu_{ij}^R(1)$ out of (27). We write the ratio of messages to row factors $\nu^R$ as

$$\tilde{\nu}_{ij}^R := \nu_{ij}^R(1)/\nu_{ij}^R(0) = (\mu_{ij}^C(1)/\mu_{ij}^C(0)) \exp(\chi_{ij}) \in \mathbb{R}^+ \,. \tag{29}$$

We symmetrically apply (27)–(29) to the column (i.e. $C$) messages $\tilde{\mu}_{ij}^C$ and $\tilde{\nu}_{ij}^C$. As is common in binary graphs, we summarize the entire message passing update scheme in terms of message ratios:

$$\tilde{\mu}_{ij}^R = \sum_{l=0}^{N_Z} \tilde{\nu}_{il}^R - \tilde{\nu}_{ij}^R \,, \quad \tilde{\nu}_{ij}^R = \frac{\exp(\chi_{ij})}{\tilde{\mu}_{ij}^C} \,, \quad \tilde{\mu}_{ij}^C = \sum_{l=0}^{N_T} \tilde{\nu}_{lj}^C - \tilde{\nu}_{ij}^C \,, \quad \tilde{\nu}_{ij}^C = \frac{\exp(\chi_{ij})}{\tilde{\mu}_{ij}^R} \,. \tag{30}$$

Finally, we compute the marginal distributions $\mathbb{E}[A_{ij}]$ by normalizing the product of the incoming messages to each variable: $\mathbb{E}[A_{ij}] = P(A_{ij} = 1) = \sigma(\chi_{ij} - \log \tilde{\mu}_{ij}^R - \log \tilde{\mu}_{ij}^C)$.

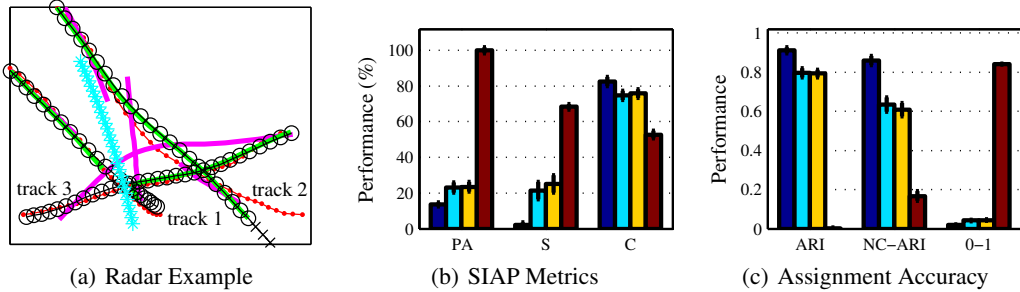

| (a) Radar Example | (b) SIAP Metrics | (c) Assignment Accuracy |

Figure 2: **Left:** The output of the trackers on the radar example. We show the true trajectories (red ·), 2D MHT (solid magenta), 3D MHT (solid green), and OMGP (cyan *). The state estimates for the VB tracker when active (black ○) and dormant (black ×) are shown, where a $\geq 90\%$ threshold on the meta-state **s** is used to deem a track active for plotting. **Center:** SIAP metrics for $N = 100$ realizations of the scenario on the left with 95% error bars. We show positional accuracy (i.e. RMSE) (PA, lower better), spurious tracks (S, lower better), and track completeness (C, higher better). The bars are in order: VB tracker (blue), 3D MHT (cyan), 2D MHT (yellow), and OMGP (red). The PA has been rescaled relative to OMGP so all metrics are in %. **Right:** Same as center but looking at assignment accuracy on ARI (higher better), no clutter (NC) ARI (higher better), and 0-1 loss (lower better) for classifying measurements as clutter.

## 5 Radar Tracking Example

We borrow the radar tracking example of the OMGP paper [18]. We have made the example more realistic by adding clutter $\lambda = 8$ and missed detections $P_D = 0.5$, which were omitted in [18]; and also used $N = 100$ realizations to get confidence intervals on the results. We also compare with the 2D and 3D (i.e. multi-frame) MHT trackers as a baseline as they are the most widely used methods in practice. The OMGP requires the number of tracks $N_T$ to be specified in advance, so we provided it with the true number of tracks, which should have given it an extra advantage. The trackers were evaluated using the SIAP metrics, which are the standard evaluation metrics in the field [7]. We also use the adjusted *Rand index* (ARI) [13] to compare the accuracy of the assignments made by the algorithms; the "no clutter" ARI (which ignores clutter) and the 0-1 loss for classifying measurements as clutter also serve as assignment metrics.

In Figure 2(a) both OMGP and 2D MHT miss the real tracks and create spurious tracks from clutter measurements. The 3D MHT does better, but misses the western portion of track 3 and makes a swap between track 1 and 3 at their intersection. By contrast, the VB tracker gets the scenario almost perfect, except for a small bit of the southern portion of track 2. In that area, VB designates the track as dormant, acknowledging that the associated measurements are likely clutter. This replaces the notion of a "confirmed" track in the standard tracking literature with a model-based method, and demonstrates the advantages of using a principled and model-based paradigm for the track management problem. This is quantitatively shown over repeated trials in Figure 2(b) in terms of positional error; even more striking are illustrations of the near lack of spurious tracks in VB and much higher completeness than the competing methods. We also show that the assignments are much more accurate in Figure 2(c). To check the statistical significance of our results we used a paired t-test to compare the difference between VB and the second best method, the 3D MHT. Both the SIAP and assignment metrics all have $p \leq 10^{-4}$.

## 6 Real Data: Video Tracking in Sports

We use the VS-PETS 2003 soccer player data set as a real data example to validate our method. The data set is a 2500 frame video of players moving around a soccer field, with annotated ground truth; the variety of player interactions make it a challenging test case for multi-object tracking algorithms. To demonstrate the robustness of our tracker to correct a detector provided minimal training examples, we used multi-scale histogram of oriented gradients (HOG) features from 50 positive and 50 negative examples of soccer players to train a sliding window support vector machine (SVM) [23]. HOG features have been shown to work particularly well for pedestrian detection on the Caltech and INRIA data sets, and thus used for this example [8]. For each frame, the center of each bounding box is provided as the only input to our tracker. Despite modest detection rates from HOG-SVM, our tracker is still capable of separating clutter and dealing with missed detections.

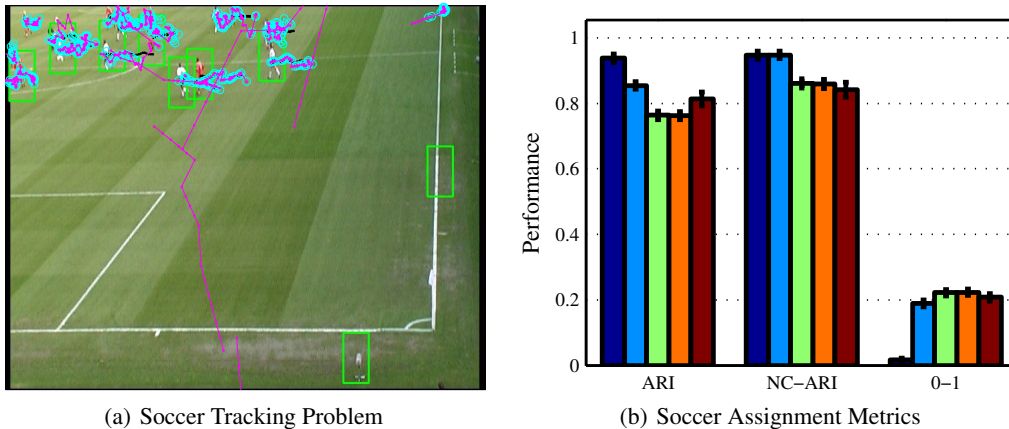

(a) Soccer Tracking Problem  (b) Soccer Assignment Metrics

Figure 3: **Left:** Example from soccer player tracking. We show the filtered state estimates of the MHT (magenta ·) and VB tracker (cyan ∘) for the last 25 frames as well as the true positions (black). The green boxes show the detection of the HOG-SVM for the current frame. **Right:** Same as Figure 2(c) but for the soccer data. Methods in order: VB-DP (dark blue), VB (light blue), 3D MHT (green), 2D MHT (orange), and OMGP (red). Soccer data source: http://www.cvg.rdg.ac.uk/slides/pets.html.

We modeled player motion using (4) with $\mathbf{F}$ and $\mathbf{Q}$ derived from an NCV model [1, Ch. 1.5]. The parameters for the NCV, $\mathbf{R}$, $P_D$, $\lambda$, and the track meta-state parameters were trained by optimizing the variational lower bound $\mathcal{L}_\beta$ on the first 1000 frames, although the algorithm did not appear sensitive to these parameters. We additionally show an extension to the VB tracker with nonparametric *clutter map* learning; we learned the clutter map by passing the training measurements into a VB Dirichlet process (DP) mixture [5] with their probability of being clutter under $q(\mathbf{A})$ as weights. The resulting posterior predictive distribution served as $p_0$ in the test phase; we refer to this method as the VB-DP tracker. We split the remainder of the data into 70 sequences of $K = 20$ frames for a test set. Due to the nature of this example, we evaluate the batch accuracy of assigning boxes to the correct players. This demonstrates the utility of our algorithm for building a database of player images for later processing and other applications. In Figure 3(b) we show the ARI and related assignment metrics for VB-DP, VB, 2D MHT, 3D MHT, and OMGP. Note that the ARI only evaluates the accuracy of the MAP assignment estimate of VB; VB additionally provides uncertainty estimates on the assignments, unlike the MHT. VB manages to increase the no clutter ARI to $0.95 \pm 0.01$ from $0.86 \pm 0.01$ for 3D MHT; and decrease the 0-1 clutter loss to $0.18 \pm 0.01$ from $0.21 \pm 0.01$ for OMGP. Using the nonparametric clutter map lowered the 0-1 loss to $0.016 \pm 0.005$ and increased the ARI to $0.94 \pm 0.01$ (vs. $0.76 \pm 0.01$ for the 2D and 3D MHT) as the VB-DP tracker knew certain areas, such as the post in the lower right, were more prone to clutter. As in the radar example the VB vs. MHT and VB vs. OMGP improvements are significant at $p \leq 10^{-4}$. The poor NC-ARI of OMGP is likely due to its lack of framing constraints, ignoring prior information on the assignments.

Furthermore, in Figure 3(a) we plot filtered state estimates for the (non-DP) VB tracker; we again use the $\geq 90\%$ meta-state threshold as a "confirmed track." We see that the MHT is tricked by the various false detections from HOG-SVM and has spurious tracks across the field; the VB tracker "introspectively" knows when a track is unlikely to be real. While both the MHT and VB detect the referee in the upper right of the frame, the VB tracker quickly sets this track to dormant when he leaves the frame. The MHT temporarily extrapolates the track into the field before destroying it.

## 7  Conclusions

The model-based manner of handling the track management problem shows clear advantages and may be the path forward for the field, which can clearly benefit from algorithms that eliminate arbitrary tuning parameters. Our method may be desirable even in tracking scenarios under which a full posterior does not confer advantages over a point estimate. We improve accuracy and reduce the exponential cost of the MAP approach to linear, which is a result of the induced factorizations of (13). We have also incorporated the often neglected framing constraints into our variational algorithm, which fits nicely with loopy belief propagation methods. Other areas, such as more sophisticated meta-state models, provide opportunities to extend this work into more applications of tracking and prove it as a general method and alternative to dominant approaches such as the MHT.

## Footnotes

[1]In this section we drop the frame index $k$ and implicitly condition on meta-states $\mathbf{S}_k$ for brevity.

[2]In most models $\text{H}_\beta[\cdot] \approx \text{H}[\cdot]$, but without proof we *always* observe $\text{H}_\beta[\cdot] \leq \text{H}[\cdot]$; so $\mathcal{L}_\beta$ is a lower bound.

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
