[Reviews · NeurIPS 2014]

Submitted by Assigned_Reviewer_11

The paper introduces a full model for tracking while allowing for multiple and varying number of hypothesis and clutter. It promises a clear notation and fast algorithms through the use of variational/Baum-Welch type inference. Experiments appear extensive and are performed on real-world data.

The key novelty of this paper is the assignment problem (aka data association). Tracking itself, as the authors acknowledge, is a well-trodden field. This maybe the reason why references are quite aged and no new developments have been cited.

The (pedantic?) reader will notice that the probability/prior for A does not involve A. While it does mention the various parameters involves the process of generating A the function does not show the user how to compute P(a_ij) or P(A) - unlike for example equation 5 in the related publication [1]. Also is a part of the normalisation of the Poisson distribution missing? Knowing this would make it easier to realise that N_Z! accounts for the random and uniform (not stated) distributions of the permutations. Unclear notation can also be found in equation 6 where the last term involves both the conditioned variable A_ij and the unconditioned variable A__ij.

References:
[1] Online Variational Approximations to non-Exponential Family Change Point Models: With Application to Radar Tracking RD Turner, S Bottone, CJ Stanek - Advances in Neural Information 2013.

Comments following reviewer discussion and author rebuttal:
Unfortunately, the authors rebuttal in itself does little to dispel my concerns. An algorithm is a also deterministic function, say of A, but not one I can use to derive the functional form of the prior. So it still stands that your likelihood was cryptic.

However, discussions with the other reviews made it clearer to me - which does also reflect poorly on the clarity of the paper's writing. Upon second reading I think the paper should be given more credit that I initially gave it.
Summary: For a reader with a background in VB and time series analysis this paper appears promising but quickly turns into frustrating read, mainly do to its ambiguous notation.

Submitted by Assigned_Reviewer_19

This paper describes a probabilistic model-based approach to tracking. The probabilistic model includes important aspects of the tracking problem (namely track management) that are often left to external heuristics and thus not included in the joint probabilistic inference (hence the term "complete" in the paper's title). In addition, by performing a richer kind of Bayesian inference (namely a mean field method with conjugate exponential family pairs as opposed to e.g. a MAP approximation) the proposed method offers a more complete representation of posterior uncertainty, improving performance and avoiding some (apparently) common tracker failure modes.

The model's construction includes Markov structure (namely in the (conditionally) linear dynamical systems (X_t,Y_t) and the Markov structure on the meta-states (S_t), shown in Figure 1) that allows the corresponding mean field updates to be carried out efficiently, requiring time only linear in the sequence (or sliding window) length as opposed to the cubic complexity in a recent Gaussian Process-based method (though at the cost of a somewhat less rich class of dynamical models for the tracks themselves). Finally, to make efficient another update required in the mean field algorithm, this paper introduces (or at least clarifies, c.f. line 117) a conjugate assignment prior (CAP) on the assignment matrices A and shows that, while some mean field computations with such variational factors are likely intractable to perform exactly (lines 193-201), a loopy belief propagation update (Section 4) can be used to estimate the expectation required to perform the mean field updates (and, possibly, to compute a lower bound on the intended mean field variational lower bound on the model evidence, c.f. 288-290 and 323). The CAP's definition is naturally related to the prior used in corresponding MAP methods (Section 2) and the choice of method for approximating the expectation is naturally related to other variational approximations (namely the Bethe entropy approximation) and to the use of similar belief propagation algorithms for the matrix permanent problem.

This is a high-quality paper. The writing is extremely clear and the explanations are very thorough. The problem, model, and approximate inference algorithms are very well motivated, and the choices made all sound reasonable.

I'm not familiar with the literature on trackers, and so my estimates of the originality and significance of this work may be both noisy and biased upwards (affecting my confidence score below). That also means I'm not in a good position to be critical of the experiments, which certainly look reasonable from my naive perspective. However, given the summary of prior work provided in Section 1, this work sounds like a clear step forward, with originality both in its more thorough Bayesian approach as well as its model and inference algorithm choices, which involve nice (if not "major") technical contributions. It may not be a "major impact" paper (I wish the impact score weren't binary...) in that its direct impact may be limited to work on trackers. But it may have a significant impact for those interested in Bayesian tracking methods (hard for me to judge) and, perhaps more importantly, I think many others in the broader NIPS community would enjoy reading this paper as much as I did.

Here are some miscellaneous comments and questions:
- I don't like using "vec" around line 175; instead, one can just define an inner product on matrices using trace.
- As a personal pet peeve, "variatonal Bayes" is somewhat less descriptive than "a mean field variational approximation to the posterior". It might be nice to include the phrase "mean field" at least once in the paper.
- Line 42, is "a complex discrete optimization task" just bipartite matching? If so, it might be nice to name.
- Around 210-212, you might consider saying "local and global" or "extensive and intensive" since those phrases are sometimes used in the literature.
- Around 164-165, the phrase "we must to compute" probably wasn't meant to include the word "to".
- On lines 96-99 and again on 426-428 you say it's surprising that by using mean field instead of MAP the (update) complexity is reduced from exponential to linear. However, since MAP can in many cases be written as mean field where some factors are constrained to be delta functions, it's not clear to me where the gain is coming from. On lines 44-45 you imply the exponential complexity comes from considering multiple frames; is the reduction in complexity really from the imposed Markov structure (namely on the s variables) which makes the assignment matrices conditionally iid? If so, it's not very surprising. If not, anything you can say to clarify the issue would be appreciated!
Summary: This paper and the work behind it are clear and thorough. While it may be of primary interest to those working on tracking methods, it is generally interesting to anyone interested in probabilistic graphical models and approximate inference.

Submitted by Assigned_Reviewer_33

The paper presents a tracking algorithm based on variational Bayes (VB). This work seems to be a natural extension of [18] and in my opinion its main novelty is the Bethe approximation in VB. Using the Bethe approximation is interesting and seems to significantly reduce the computational complexity of the problem. The experiments seem to be well executed.

The main drawback of the paper is its writing. The equations considerably obscure the ideas of the paper. This significantly affects the readability of the paper and I am afraid that in its current form it will not be able to make an impact.
Summary: The paper presents interesting connections between Bethe entropy approximation and variational Bayes for a tracking task. However, the readability of the paper is a real concern in this case.
Author Feedback
Author rebuttal: We thank the reviewers for their useful comments.

Although reviewer 11 is very critical the review is quite short and doesn't leave much to respond to.

Most of the review (besides restating the abstract) relates to equation 2: "The (pedantic?) reader will notice that the probability/prior for A [eqn. 2] does not involve A." The *very next* sentence in the paper states, "Note that the detections d, NZ, and clutter measurement count Nc are deterministic functions of A", while the prior on A is explicitly parameterized by d, NZ, and Nc. Perhaps we should write d, NZ and Nc explicitly as d(A), NZ(A), and Nc(A) to make this point more clear. The comment that "... the function does not show the user how to compute P(a_ij) or P(A) - unlike for example equation 5 in the related publication [1]" is confusing since the terms in equation 5 of [1] that address P(A) are also explicitly written into equation (2) of this paper. Perhaps some supplemental material explicitly deriving each of the terms in equation (2) would help.

Reviewer 33 raises some concerns over the density of the equations. However, reviewer 19 states "The writing is extremely clear and the explanations are very thorough". So, it is not clear if anything must be changed in that dimension.

Reviewer 19's asks an insightful question with "However, since MAP can in many cases be written as mean field where some factors are constrained to be delta functions, it's not clear to me where the [computational] gain is coming from." We think the best intuition for the gain is to consider the induced factorization in the q(A_1:K) posterior in eqn. (13). Although, you can setup MAP as a VB problem, the MAP variational constraints do not yield the same induced factorizations as in (13) and therefore don't yield the same computational benefit.